# Collision-Avoidance Decision System for Inland Ships Based on Velocity Obstacle Algorithms

**Guangyu Zhang** [1], **Yan Wang** [1], **Jian Liu** [2], **Wei Cai** [2] **and Hongbo Wang** [1,*]

1   State Key Laboratory on Integrated Optoelectronics, College of Electronic Science and Engineering, Jilin University, Changchun 130012, China; guangyu19@mails.jlu.edu.cn (G.Z.); yanwang20@mails.jlu.edu.cn (Y.W.)
2   Tianjin Navigation Instrument Research Institute, Jiujiang 320007, China; winster@live.cn (J.L.); cw6324235@163.com (W.C.)
*   Correspondence: wang_hongbo@jlu.edu.cn; Tel.: +86-0431-8516-8270

**Abstract:** Due to the complex hydrology and narrow channels of inland rivers, ship collision accidents occur frequently. The traditional collision-avoidance algorithms are often aimed at sea areas, and not often at inland rivers. To solve the problem of inland-ship collision avoidance, this paper proposes an inland-ship collision-avoidance decision system based on the velocity obstacle algorithm. The system is designed to assist ships in achieving independent collision-avoidance operations under the limitation of maneuverability while meeting inland-ship collision-avoidance regulations. First, the paper improves the Maneuvering Modeling Group (MMG) model suitable for inland rivers. Then, it improves velocity obstacle algorithms based on the dynamic ship domain, which can deal with different obstacles and three encounter situations (head-on, crossing, and overtaking situations). In addition, this paper proposes a method to deal with close-quarters situations. Finally, the simulation environment built by MATLAB software is used to simulate the collision avoidance of inland ships against different obstacles under different situations with a decision-making time of less than 0.1 s. Through the analysis of the simulation results, the effectiveness and practicability of the system are verified, which can provide reasonable collision-avoidance decisions for inland ships.

**Keywords:** inland ship; collision avoidance; velocity obstacle; MMG model

## 1. Introduction

As one of the important forms of transportation, inland navigation plays an essential role in the transportation system. Nowadays, with the gradual increase in the number of inland ships, the navigation density is also increasing. Due to the bending and complex hydrology of inland rivers, ship collision accidents occur frequently, and inland navigation safety is becoming more serious. In addition, in recent years, more and more inland unmanned surface vehicles (USVs) have been used. Giordano et al. have described a prototype of a marine drone optimized for very shallow water, which enables bathymetric surveys to be performed in areas that are not feasible for traditional boats [1]. Stateczny et al. completed the study on the positioning accuracy of Global Navigation Satellite System (GNSS)/Inertial Navigation Systems (INS) systems supported by Differential Global Positioning System (DGPS) and Real Time Kinematic (RTK) receivers for hydrographic surveys by using a USV [2]. Nikolakopoulos et al. analyzed USV data to investigate beach rock formations [3]. Specht et al. proposed the concept of an innovative autonomous unmanned system for shallow water depth monitoring [4]. Concerning the problem in USVs' navigation mode without the satellite navigation system, Wang et al. proposed the multi-sensor integrated navigation method of Inertial Navigation System (INS)/Celestial Navigation System (CNS)/Doppler Velocity Log (DVL) using Adaptive Information Sharing Factor Federated Filter (AISFF) [5]. It can be seen that inland USVs also have the problem of collision avoidance in restricted waters or shallow waters. To solve this problem, this paper

improved the velocity obstacle algorithm and proposed an inland-ship collision-avoidance decision-making system based on the velocity obstacle algorithm.

The traditional ship collision-avoidance algorithms, such as Artificial Potential Field [6], Genetic Algorithms [7,8], Annealing Algorithms [9], and Particle Swarm Optimization [10,11], mainly focus on sea areas. Artificial Potential Field based on the virtual physical potential field applies to a sea area with few obstacles [12], but not to inland rivers because of the rudder effect and complex navigation conditions of inland ships. A Genetic Algorithm with one-dimensional coding has achieved good results in the curved channel [13]. However, it has limitations in dealing with collision avoidance among inland ships, and it is also unable to deal with collision avoidance in close-quarters situations. At the same time, there are new intelligent algorithms based on artificial intelligence [14], data processing [15], and scene prediction [16]. However, the collision-avoidance algorithm suitable for inland rivers remains to be studied.

The inland-ship collision-avoidance algorithm requires reliability, real-time performance, and stability and can be applied to a variety of encounter scenarios. Sun [17] suggests that when two ships meet in a narrow channel and reach the beam of another ship, they are considered to have passed safely. Even if one of them takes unexpected actions, it will not lead to a collision. Zhu et al. [18] suggested collision time and collision distance factors and used the double detection window to avoid static obstacles. Ahn et al. [19] designed a fuzzy neural algorithm based on neural network and fuzzy set theory and established a ship collision-avoidance expert system. Ugur et al. [20] established a collision-avoidance decision support system based on a neural network, which can predict the ship's position after three minutes and broadcast in combination with Vessel Traffic Service (VTS) to prevent ship collision. He et al. [21] proposed the judgment process of the give-way ship stage and the calculation process of the close-quarters situation and close-quarters formation point. Mao et al. [22] combined TCPA with fuzzy logic as a multi-obstacle collision-avoidance strategy and considered the two online obstacles with the highest urgency in each update round. Liu [23] adopted an optimized wolf colony search algorithm and an optimized bacterial foraging algorithm to research ship intelligent automatic collision-avoidance strategies. Alonso-Mora et al. [24] proposed the concept of collaborative collision avoidance (CCA) to prevent reciprocal dances. Based on the quaternion ship domain, Li et al. [25] combined a fast non-dominated sorting Genetic Algorithm (NSGA-II) with a decomposition-based multi-objective evolutionary algorithm (MOEA/D) to calculate the ship collision-avoidance route. As a result, they realized a multi-objective intelligent collision-avoidance algorithm. Song et al. [26] designed the eccentric expansion circle to discover the ship collision-avoidance algorithm meeting International Regulations for Preventing Collisions at Sea (COLREGs). Du et al. [27] and Gu et al. [28] improved the A* algorithm and named it 'label- A* algorithm' (LAA). They combined the trajectory unit with LAA to achieve USV motion planning in restricted waters.

Fiorini proposed the velocity obstacle algorithm (VO) in 1998 [29], which was initially used to solve the problem of autonomous obstacle avoidance of mobile robots. Later, many different versions were derived according to the needs of the scene and applied to many fields, such as unmanned aerial vehicles (UAVs), and unmanned ground vehicles (UGVs). In 2014, the VO algorithm was used for the first time for collision avoidance of USVs. Scholars constructed velocity obstacle areas according to the size of USVs and avoidance regulations [30]. In recent years, with the popularization of the concept of the intelligent ship, the VO algorithm has been gradually applied to ship collision-avoidance strategies in sea areas [31]. The principle of the VO algorithm for ship collision avoidance is shown in Figure 1.

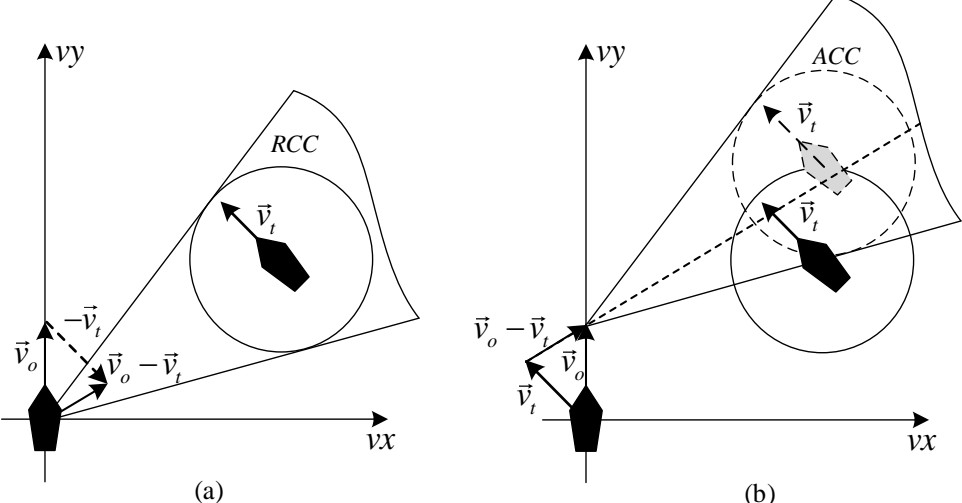

**Figure 1.** Velocity obstacle algorithm: (**a**) *RCC* and (**b**) *ACC*.

In Figure 1, the *vx*-axis points east, the *vy*-axis points north, $\vec{v}_o$ is the velocity vector of the own ship, $\vec{v}_t$ is the velocity vector of the target ship. *RCC* represents the relative collision cone, and the composition of the resultant velocity is $\vec{v}_o + (-\vec{v}_t)$. Figure 1a shows that the resulting velocity is within the *RCC*, and a collision risk exists if the ship continues to sail, so collision avoidance should be considered. If the target ship's velocity $\vec{v}_t$ is translated to the origin, then take the end of $\vec{v}_t$ as the starting point to translate the *RCC*. As a result, the new cone area will be an absolute collision cone (*ACC*). Figure 1b shows that the starting point of the resulting velocity is at the end of $\vec{v}_t$, and the composition is $\vec{v}_o - \vec{v}_t$. There will be a collision risk if the ship continues to sail. The conclusion is consistent with Figure 1a. Therefore, consistent results will be obtained using either *RCC* or *ACC* for judgment. The above findings are expressed as follows:

$$ACC = RCC \oplus \vec{v}_t \tag{1}$$

where $\oplus$ represents the Minkowski sum.

Previous collision-avoidance algorithms, such as Ant Colony Algorithms, Genetic Algorithms, and Particle Swarm Optimization, often need to carry out a large number of iterations in advance to obtain the results. The running time also increases with the increase of the number of ships and the number of iterations, generally more than 5 s. Therefore, these algorithms are more suitable for global path planning, with the advantage that multiple obstacle avoidance paths can be obtained. However, in case of a close-quarters situation, once the own ship is unable to coordinate with the other ship immediately, the collision-avoidance opportunity will be delayed, resulting in the accident. Based on geometric principles, the VO algorithm has a small amount of calculation and uniform results, which meets the basic requirements of the inland-ship collision-avoidance algorithm. Notably, the ship can make decisions independently. This allows the ship to receive immediate collision-avoidance instructions in the case of a close-quarters situation. In addition, in inland rivers, multi-ship encounter situations often occur when the number of ships is more than ten. For this problem, some scholars have used the VO algorithm to complete the obstacle avoidance simulation test of 1000 agents [32], which is difficult for other algorithms. Therefore, this paper will improve the VO algorithm to realize the inland-ship collision-avoidance decision-making system.

In this paper, the authors propose an intelligent and effective method based on an improved VO algorithm to solve the collision-avoidance problem of multiple ships in inland rivers. The contributions are as follows:

- The mathematical model of inland ships is established, including a maneuverability motion model for inland ships, an improved ship domain, and a collision risk model. The ship model is modified by the Maneuvering Modeling Group (MMG) model considering the force of inland ships. The model considers the effects of wind, current, shallow water, and banks and recalculates the hydrodynamic parameters. Finally, the ship model is verified and compared with the real ship data through the turning and Z-type tests.
- The VO algorithm is improved as a collision-avoidance algorithm that includes the following steps. For collision avoidance of static obstacles, the classified obstacles are modeled using the grid method. For dynamic obstacles, the VO cone is constructed by giving up the circle and selecting an elliptical ship domain. For multi-ship encounter situations, the VO area is redivided considering safety distance and potential hazards. Finally, for close-quarters situations, a speed buffer zone is established.
- Simulation experiments were performed to simulate inland-ship avoidance: static obstacles that include shoals and reefs, and three dynamic obstacle encounter situations: head-on, overtaking, and crossing. On this basis, the paper verifies the collision avoidance of multi-ship situations in inland channels. In addition, the collision avoidance in close-quarters situations is verified. The simulation results show that each ship can fulfill independent collision-avoidance operations according to the inland-river collision-avoidance regulations. Compared with previous ship collision-avoidance algorithms, the operation speed is greatly improved. Furthermore, the collision-avoidance instructions can be given in real time, meeting the actual requirements.

The rest of the paper is organized as follows: Section 2 describes the mathematical model of the inland ship and associated risk functions. Section 3 introduces the principles of the improved VO algorithm. Section 4 shows the simulation results of multi-ship collision-avoidance scenarios. Finally, Section 5 presents the conclusions.

## 2. Mathematical Model of Inland Ship

### 2.1. Avoidance Regulations and Inland-Ship Domain

In this paper, the collision-avoidance regulations of inland ships are designed according to COLREGs. The encounter situations are also divided into three types: head-on, overtaking, and crossing. The judgment method of the three encounter situations is as follows:

$$(\theta_t \leq 5° \text{ or } \theta_t \geq 355°) \text{ and } |180° - |\varphi_t - \varphi_o|| \leq 5° \tag{2}$$

$$112.5° \leq \theta_t \leq 247.5° \tag{3}$$

$$5° < \theta_t < 112.5° \text{ or } 247.5° < \theta_t < 355° \tag{4}$$

where $\theta_t$ represents relative bearing angle. $\varphi_t$ and $\varphi_o$ represent the heading angle of the target ship and own ship, respectively. Equations (2)–(4) are the judgment basis of head-on, overtaking, and crossing. The corresponding ship avoidance behaviors in the three encounter situations are shown in Figure 2.

In Figure 2, the two ships should pass on the port side in a head-on situation. In the overtaking situation, the overtaking ship is a give-way ship, and the overtaking ship should meet the overtaken ship on the starboard side. In the crossing situation, if there is a target ship coming from the starboard angle of 5° to 112.5°, the own ship is a give-way ship. Then, the own ship shall meet on the port side and pass the stern of the ship that has given way. If there is a ship from the port angle of 5° to 112.5°, the target ship shall be considered a give-way ship. It shall also meet the own ship on the port side and pass the stern of the own ship.

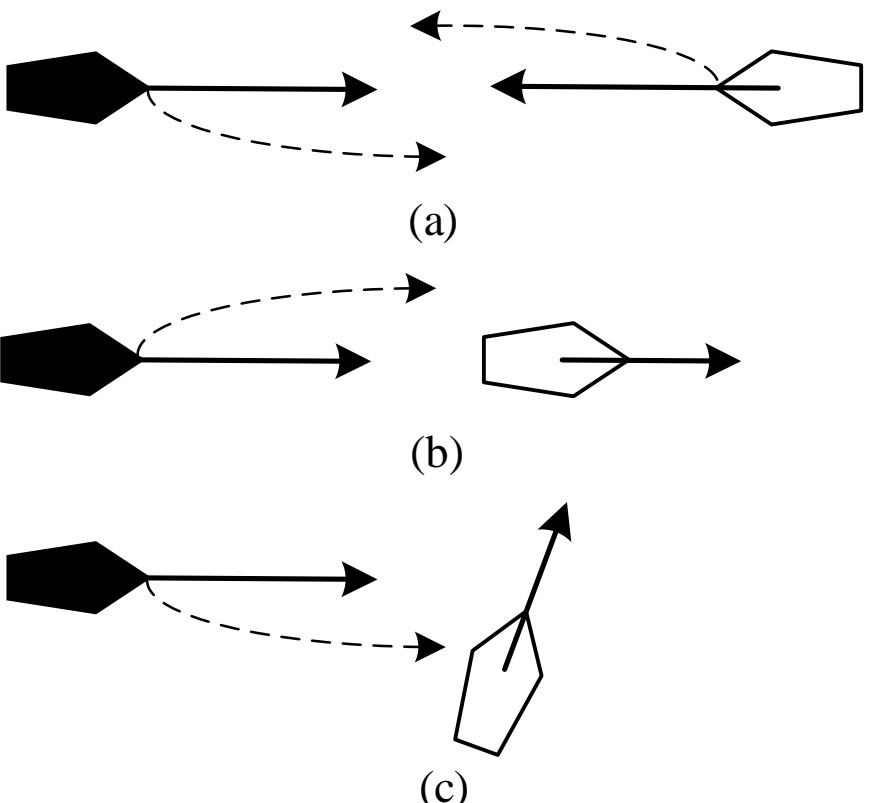

**Figure 2.** Inland-ship encounter situations: (**a**) head-on, (**b**) overtaking, and (**c**) crossing.

The ship domain is the smallest water area where ships can navigate safely. If a target ship enters the ship domain, it will form a close-quarters situation. At this time, if necessary, both ships should actively avoid or even abandon the inland-river collision-avoidance regulations. Traditional ship domains include the Goodwin, Coldwell, Fujii [33], and Quaternion models [34]. Combining the above models with the inland-river environment, this paper designs the inland-ship domain shown in Figure 3.

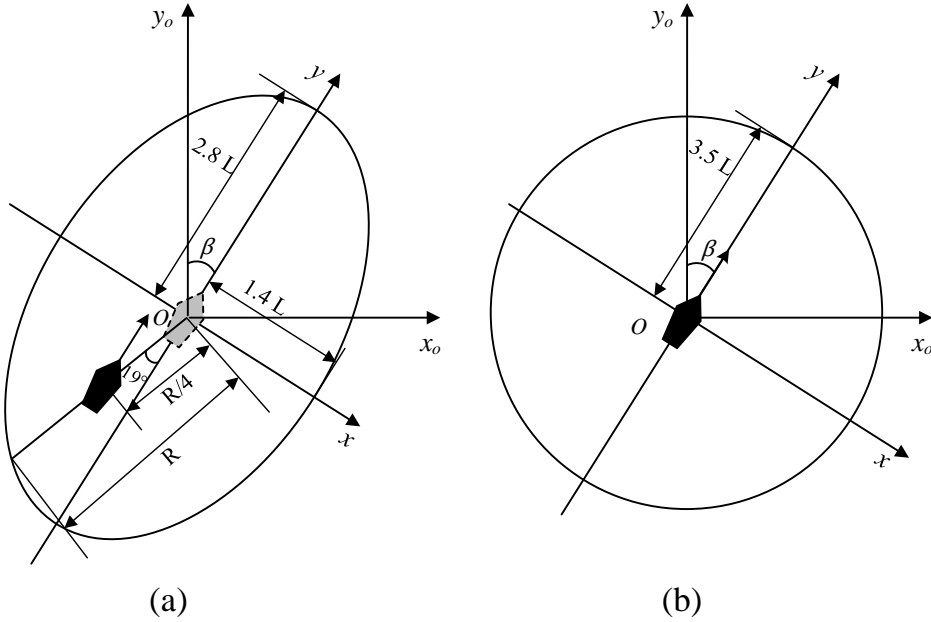

**Figure 3.** Inland-ship domain for: (**a**) dynamic obstacles, (**b**) static obstacles.

In Figure 3, L represents ship length and $\beta$ represents drift angle. Figure 3a shows the ship domain dealing with dynamic obstacles. The inland waterways are narrow and long. Based on ship-type comparisons and weighted averages, the semi-major axis of the ship domain is positioned at 2.8 L and the semi-minor axis at 1.4 L. According to the Goodwin, Coldwell, and Quaternion ship domains, the ship's position is taken on the left side. Then, according to the long oval ship domain of the Fujii model, the ship's position is moved downward, and the elliptical ship domain is established at the gray virtual ship. This meets the inland narrow channel, puts the ship in the safest position in the ship domain, and meets the requirements of avoidance regulations. In Figure 3b, the ship is still in the center since static obstacles do not need to consider avoidance regulations. The circular ship domain is designed to consider inland-river banks. The radius of the circle is 3.5 L. With such improvements, the inland ship domain will change in response to environmental changes. The dynamic design makes collision avoidance more flexible.

### 2.2. Ship Motion Model

In addition to the rudder force and propeller force, inland ships need to consider the forces generated by external wind, currents, bank effects, and shallow-water effects. Therefore, to truly reflect the navigation trajectory of a ship, accurate ship models are needed for prediction. Inland-ship models with three degrees of freedom can be selected, including surging, swaying, and yawing. The hydrodynamic forces or moments corresponding to the three degrees of freedom are represented by $X$, $Y$, and $N$, respectively. The ship-maneuvering model is shown in Figure 4.

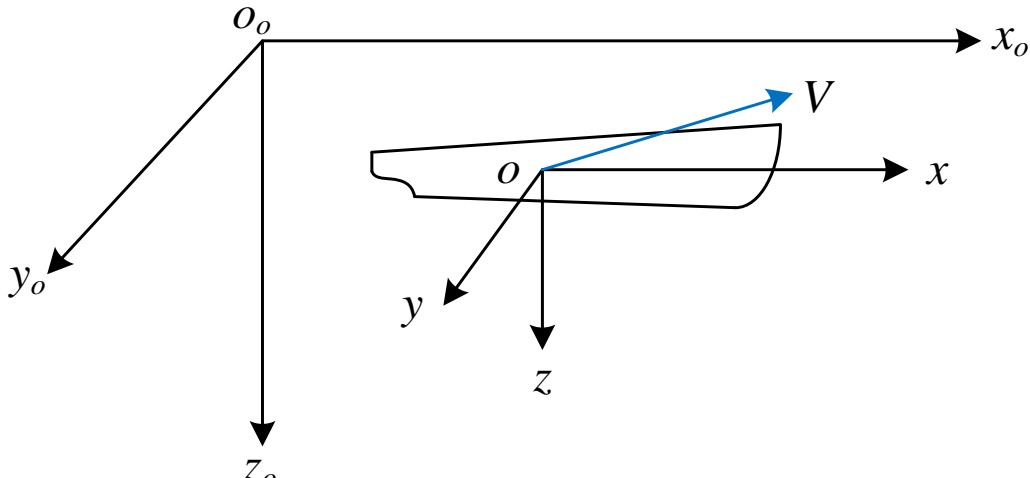

**Figure 4.** Ship-maneuverability motion coordinate system.

In Figure 4, the coordinate system consists of two right-hand coordinate systems, in which $o_o - x_o y_o z_o$ is the inertial coordinate system fixed on the earth's surface, which is used to mark the geographical position of the ship. The $x_o$-axis points north, the $y_o$-axis points east, and the $z_o$-axis points to the earth's center. $o - xyz$ is a motion coordinate system with its origin fixed at the ship's center of gravity, which is used to record the motion state of the ship. The $x$-axis points to the bow, the $y$-axis points to the starboard, and the $z$-axis is perpendicular to the waterplane. The horizontal view of the coordinate system is shown in Figure 5, and physical parameters are shown in Table 1.

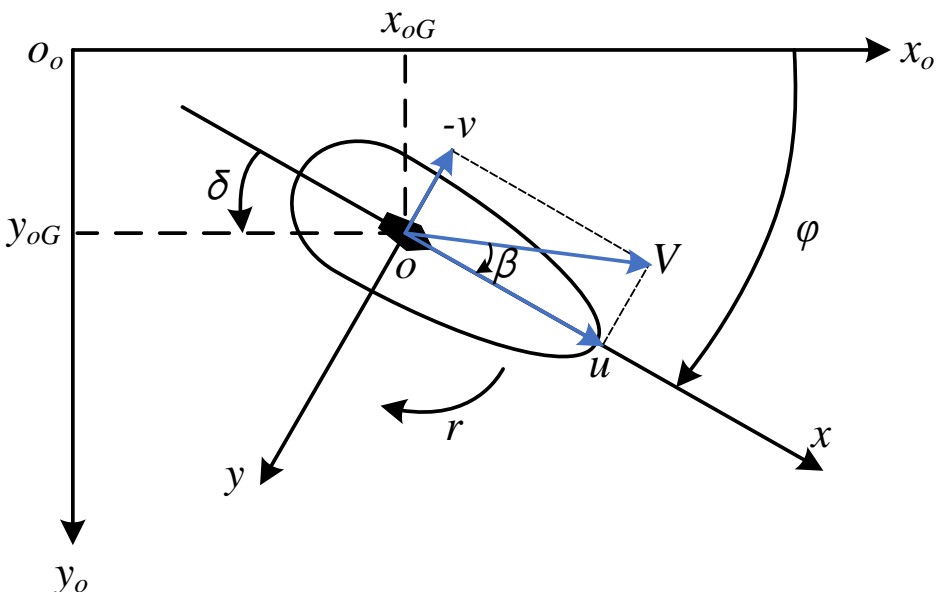

**Figure 5.** The horizontal view of the ship's maneuverability motion coordinate system.

**Table 1.** Coordinate system parameters.

| Parameters | Meanings | Unit |
|:---:|:---:|:---:|
| $o_o - x_o y_o z_o$ | fixed coordinate system | m |
| $o - xyz$ | motion coordinate system | m |
| $u$ | longitudinal velocity | m/s |
| $v$ | lateral velocity | m/s |
| $r$ | yawing angular velocity | rad/s |
| $\beta$ | drift angle | rad |
| $\varphi$ | heading angle | rad |
| $\delta$ | rudder angle | rad |
| $V$ | ship speed | m/s |

The MMG model was proposed by the Japanese Towing Tank Conference Commission (JTTC) in 1972. The ship model is established by considering the separate hydrodynamic forces or moments on the bare hull, propeller, and rudder. In this paper, we extended it to inland rivers. The improved modeling idea is as follows:

$$\left.\begin{array}{l} f(X) = X_H + X_P + X_R + X_E \\ f(Y) = Y_H + Y_P + Y_R + Y_E \\ f(N) = N_H + N_P + N_R + N_E \end{array}\right\} \tag{5}$$

where subscripts *H*, *P*, and *R* represent the force or moment from the bare hull, propeller, and rudder, respectively. Subscript *E* represents the environmental impact on the inland ship, such as wind, currents, and bank effects. The correction of each part of the force in shallow water is as follows:

(1)   Correction of bare-hull force

The hydrodynamic coefficients in deep water can be estimated by the Inoue and Kijima models, while they need to be corrected in shallow water [35]. The water depth correction function is as follows:

$$D_{shallow} = f(\lambda) \times D_{deep} \tag{6}$$

$$\lambda = d/h \tag{7}$$

where $D_{deep}$ represents the estimation formula of deep water, $D_{shallow}$ represents the estimation formula of shallow water, $f(\lambda)$ represents the water-depth-correction function,

$d$ represents the average ship draft, $h$ represents the water depth, and $\lambda$ represents the water-depth draft ratio.

Different hydrodynamic coefficients have different depth-correction functions and constant values, including $Y_v'$, $Y_{vv}'$, $Y_{vrr}'$, $N_v'$, and $N_r'$:

$$f(\lambda) = 1/(1-\lambda)^n - \lambda \tag{8}$$

where $n$ is a constant coefficient. For other hydrodynamic coefficients, the depth-correction function can be expressed as:

$$f(\lambda) = 1 + a_1\lambda + a_2\lambda^2 + a_3\lambda^3 \tag{9}$$

where $a_1$, $a_2$, and $a_3$ represent constant coefficients.

(2)　Correction of propeller force

In shallow water, the calculation formula of $X_P$ remains unchanged. However, the thrust deduction factor $t_P$ and wake factor $w_P$ will be affected to some extent. $1 - t_P$ will slightly decrease with the decrease in water depth. In contrast, $1 - w_P$ will significantly decrease with the reduction in water depth. Hence, the correction formula is as follows:

$$\frac{(1-t_P)_h}{(1-t_P)_\infty} = \frac{1}{1 - 0.2(d/h) + 0.7295(d/h)^2} \tag{10}$$

$$\frac{(1-w_P)_h}{(1-w_P)_\infty} = \cos\left(1.4C_b\frac{d}{h}\right) \tag{11}$$

where $C_b$ is a block coefficient.

(3)　Correction of rudder force

In shallow water, the calculation formula of $X_R$ remains unchanged. However, the flow-straightening factor $\gamma$, flow-increasing factor $a_H$, and the distance from the fluid force point to the ship's center of gravity $x_H$ will be affected. $\gamma$ will increase with the decrease in water depth and decrease after the turning point. As the water becomes shallower, $a_H$ increases and $x_H$ decreases. Therefore, the correction formula is as follows:

$$\frac{\gamma}{\gamma_\infty} = 1 + 0.0161\frac{d}{h} + 4.4222\left(\frac{d}{h}\right)^2 - 4.9825\left(\frac{d}{h}\right)^3 \tag{12}$$

$$\frac{\alpha_H}{\alpha_{H\infty}} = 1 + 0.3621\frac{d}{h} + 1.1724\left(\frac{d}{h}\right)^2 \tag{13}$$

$$\frac{x_H}{x_{H\infty}} = 1 + 0.3328\frac{d}{h} - 3.2134\left(\frac{d}{h}\right)^2 + 2.5916\left(\frac{d}{h}\right)^3 \tag{14}$$

(4)　Analysis of external environmental forces

External impacts on ships often include wind, currents, and bank effects. In general, the effects of wind forces and moments on the hull are as follows:

$$\left.\begin{array}{l} X_{wind} = \frac{1}{2}\rho_a A_f U_r^2 C_{wx}(a_R) \\ Y_{wind} = \frac{1}{2}\rho_a A_s U_r^2 C_{wy}(a_R) \\ N_{wind} = \frac{1}{2}\rho_a A_s L U_r^2 C_{wn}(a_R) \end{array}\right\} \tag{15}$$

where $\rho_a$ represents air density, $U_r$ represents relative wind speed, $A_f$ represents the positive projection area above the waterline, and $A_s$ represents the side projection area. $C_{wx}(a_R)$ and $C_{wy}(a_R)$ represent the wind-pressure coefficient, and $C_{wn}(a_R)$ represent the wind-moment coefficient, which can be determined by the Isherwood formula.

Since the influence of waves on inland ships is limited, only the impact of currents is considered. Therefore, the cross-flow formula is as follows:

$$\left.\begin{array}{l} Y_{current} = Y_v v_r + Y_r r = \rho L d V_r^2 \frac{\pi}{2}(2a + b) \\ N_{current} = N_v v_r + N_r r = \rho L^2 d V_r^2 \frac{\pi}{4}a \end{array}\right\} \tag{16}$$

where $\rho$ represents water density, $V_r$ represents the relative speed of the ship to water, $v_r$ represents the lateral velocity of the ship relative to water, $a$ and $b$ are unknown coefficients, which can be determined according to the low-aspect-ratio wing theory.

In addition, when a ship is sailing in a narrow channel, due to the fluid change caused by the hull, the flow velocity near the bank is faster, and the water pressure is lower. As a result, the bank thrust at the bow and the bank suction at the stern form a turning moment, and the ship receives bank suction. The formula for the effect of the bank force on the ship is as follows:

$$\left.\begin{array}{l} Y_S = \rho C_b B d u^2 Q \left[0.0925 + 0.327\left(\frac{d}{h}\right)^2\right] \\ N_S = -\rho C_b B d u^2 Q \left[0.0025 + 0.0755\left(\frac{d}{h}\right)^2\right] L \end{array}\right\} \tag{17}$$

where $B$ represents the width of the ship, and $Q$ represents the distance from the ship's center to the bank.

In this paper, the turning and Z-type tests verify the inland-ship model. The ship data are from the merchant ship "HUAIJI River." The scale parameters of the real ship and ship model are shown in Table 2:

**Table 2.** Main scale table of test ship and ship model.

| Parts | Ship Scale Parameters | | Real Ship | Ship Model |
|---|---|---|---|---|
| hull | ship length | $L$ (m) | 114.0 | 2.00 |
| | ship width | $B$ (m) | 20.5 | 0.351 |
| | designed draft | $d$ (m) | 6.9 | 0.118 |
| | block coefficient | $C_b$ | 0.653 | |
| | prismatic coefficient | $C_p$ | 0.692 | |
| | displacement volume | $\bigtriangledown$(m³) | 10805.7 | 0.0544 |
| rudder | rudder area | $A_R$ (m²) | 17.95 | 0.005275 |
| | rudder height | $H_R$ (m) | 5.49 | 0.094 |
| | rudder area ratio | $\lambda_R$ (m) | 1.68 | |
| | rudder area ratio | $A_R/Ld$ | 0.024 | |
| paddle | propeller diameter | $H_P$ (m) | 4.5 | 0.077 |
| | pitch ratio | $P/D_P$ | 0.89 | |
| | disc area ratio | $\theta$ | 0.71 | |
| | rudder quadrant | $Z_R$ | 4 | |

To test the trajectory of an inland-ship model in shallow water, the turning test simulation was performed for four draft ratios. The ship was operated with the right full rudder at a 35° rudder angle. The simulation results obtained by MATLAB are shown in Figure 6 and Table 3.

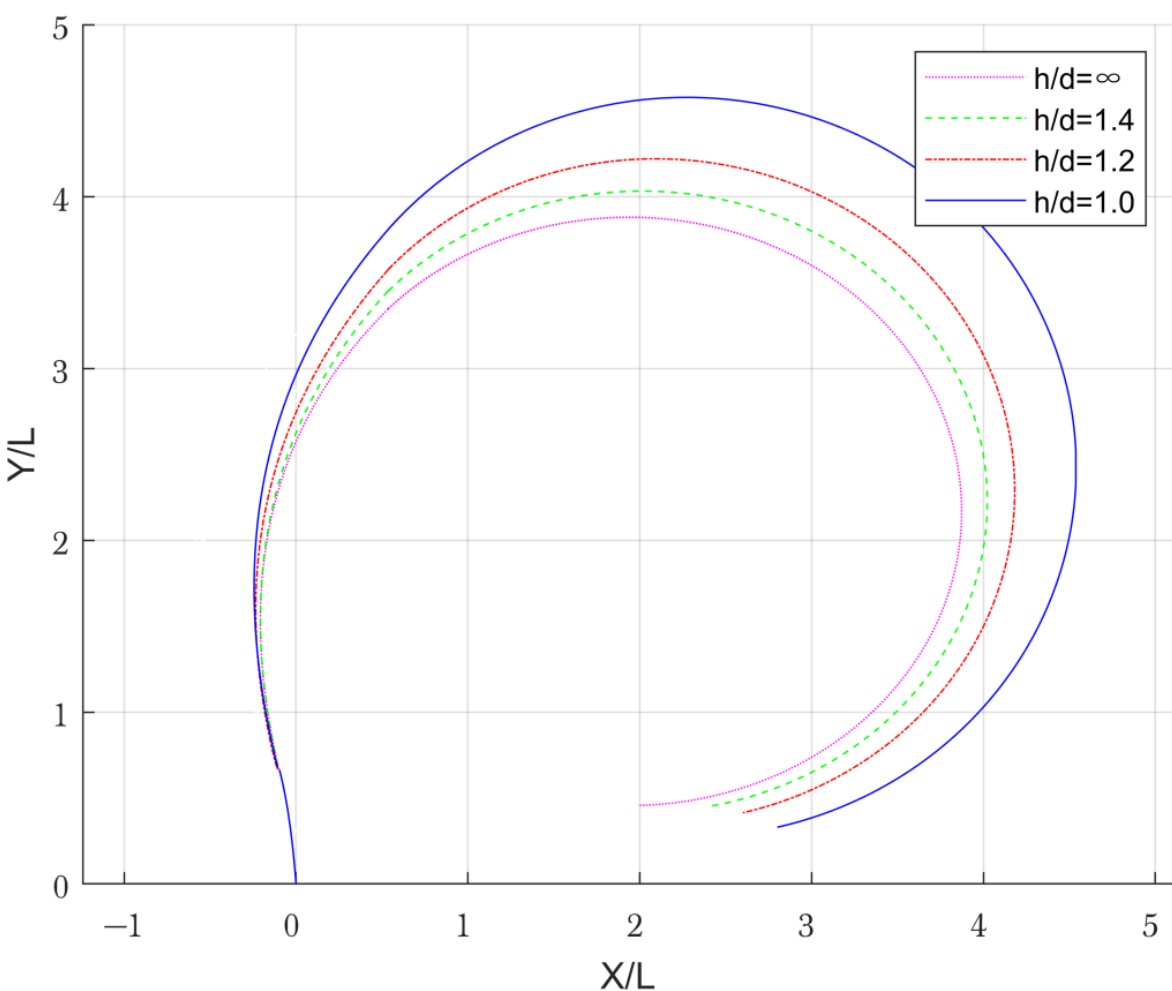

**Figure 6.** Turning test trajectory.

**Table 3.** Turning test results.

| Parameters | h/d = 1.0 | h/d = 1.2 | h/d = 1.4 | h/d=∞ |
|---|---|---|---|---|
| Initial velocity (m/s) | 6.11 | 6.11 | 6.11 | 6.11 |
| Advance (m) | 519.5 | 470.8 | 461.8 | 444.6 |
| Transfer (m) | 226.3 | 210.6 | 199.4 | 193.8 |
| Tactical diameter (m) | 518.7 | 484.5 | 456.0 | 442.3 |
| Final diameter (m) | 473.1 | 427.8 | 401.9 | 376.2 |

In Figure 6, the ship starts from the coordinate origin and turns to the right. The advance is 3.8–5.5 L, and the transfer is 1.7–2 L. The simulation results show that, when the water is infinitely deep, the tactical diameter is about 3 L. The shallow-water effect becomes more apparent with the continuous reduction of the water depth. The resistance gradually increases, increasing the difficulty of ship maneuverability. As a result, the tactical diameter gradually increases. When the draft ratio approaches 1.0, the tactical diameter of the ship is about 4 L.

Furthermore, the Z-type test is performed on the ship model. The Z-type test is an important method for detecting ship maneuverability and predicting ship trajectory. In this paper, the rudder angle is set to $\pm 10°$, and the simulation result of the right $10°/10°$ Z-type maneuver test is shown in Figure 7.

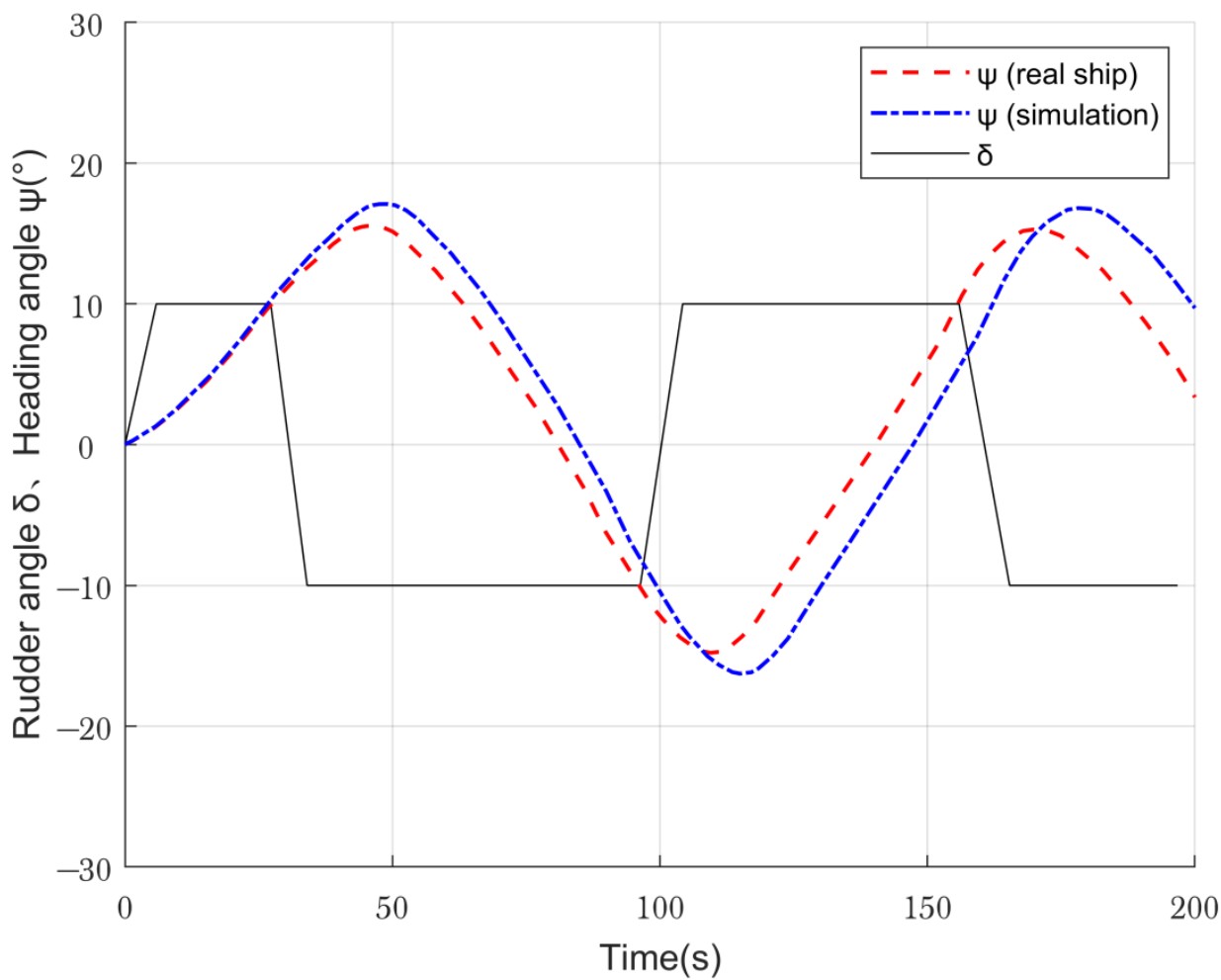

**Figure 7.** Right $10°/10°$ Z-type maneuver test.

In Figure 7, the solid line indicates the change in the rudder angle. The heading angle of the ship model is the chain line, and the dash line shows the real ship data. An angle greater than zero refers to the starboard side, and less than zero refers to the port side. Through comparison, it can be seen that within the first 100 s, the heading angle difference between the simulation data and the real ship data is about 1°, and the time difference is within 2 s. It is evident that the heading angles of the two groups of data are almost consistent with the change of rudder angle. This proves that the ship model can roughly capture the maneuvering motions in a reasonable time and is useful for the maneuvering predictions of inland ships.

### 2.3. Collision Risk Model

To prioritize collision avoidance for different obstacles, this paper adopts the fuzzy logic theory and improves it to quantify the degree of ship collision risk. The design of the fuzzy set is as follows:

$$U = [DCPA, TCPA, D, K, Q] \tag{18}$$

where *DCPA* refers to the shortest encounter distance of the ship, *TCPA* refers to the shortest encounter time, *D* refers to the distance between the two ships, *K* refers to the ship speed ratio, and *Q* refers to the distance from the ship's center of gravity to the bank. As mentioned above, inland ships are affected by bank effects. The suction generated by

the bank will affect the ship's maneuverability. The narrower the channel, the higher the collision risk. Therefore, the membership function of the channel width is as follows:

$$U(Q) = \begin{cases} 1, & 0 \leq Q \leq Q_1 \\ \frac{1}{2} - \frac{1}{2}\sin\left[\frac{\pi}{Q_2 - Q_1}\left(Q - \frac{Q_1 + Q_2}{2}\right)\right], & Q_1 < Q \leq Q_2 \\ 0, & Q_2 < Q \end{cases} \tag{19}$$

where $Q_1$ is the nearest avoidance distance from the bank, and $Q_2$ is the safe avoidance distance from the bank. Finally, the computing method of inland-ship collision risk is as follows:

$$U_{CRI} = a_{DCPA}U_{DCPA} + a_{TCPA}U_{TCPA} + a_D U_D + a_K U_K + a_Q U_Q \tag{20}$$

where $a$ is the weight, which is set to 0.32, 0.30, 0.16, 0.10, and 0.12 according to the Analytic Hierarchy Process, respectively, for the above membership degrees. Furthermore, the degree of risk of dynamic obstacles is transformed from quantitative to qualitative. The collision risk level is shown in Table 4:

**Table 4.** Collision risk level.

| Risk Level | Collision Risk | Assessment | Behavior |
|:---:|:---:|:---:|:---:|
| I | 1.00–0.81 | maximum risk | close-quarters situation |
| II | 0.80–0.61 | high risk | |
| III | 0.60–0.41 | medium risk | make avoidance decisions |
| IV | 0.40–0.21 | potential risk | free navigation stage |
| V | 0.20–0 | low risk | |

The ships with Classes IV and V may not take collision-avoidance measures, namely the free navigation stage, but those with Class IV risk levels are considered potential hazards. Medium risk indicates a collision risk between the own ship and the target ship. Hence, it is necessary to make a collision-avoidance decision for the own ship and see whether the navigation route needs to be changed according to the avoidance regulations. When the risk level is maximum or high risk, the ship has entered a close-quarters situation. Therefore, ships should take collision-avoidance decisions immediately and turn with full rudder. If necessary, they can even deviate from the avoidance regulations.

Figure 8 shows the simulation and verification of the collision-risk model.

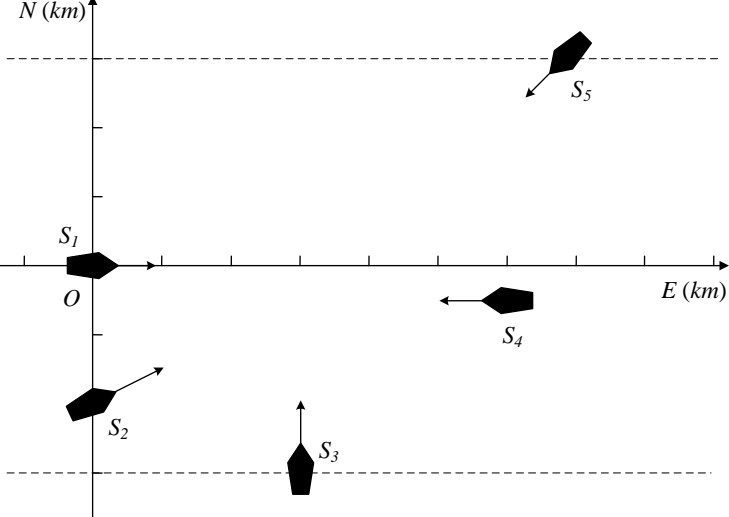

**Figure 8.** Scenario verification diagram.

In Figure 8, the unit length of the coordinate axis is 2 km. The own ship is at its origin, and there are four target ships around it, forming different encounter situations with the own ship. The risk level of the target ship is analyzed using the risk model, and the result is shown in Table 5:

**Table 5.** Collision risk level.

| No. | Position (km) | Speed (km/h) | Heading (°) | Risk | Level |
|---|---|---|---|---|---|
| S1 | [0, 0] | 18 | 90 | - | - |
| S2 | [0, −4] | 21 | 60 | 0.535 | III |
| S3 | [6, −6] | 18 | 0 | 0.577 | III |
| S4 | [12, −1] | 18 | 270 | 0.469 | III |
| S5 | [14, 6] | 18 | 225 | 0.382 | IV |

Ship S3 has the highest collision risk around S1, which is close to Class II and needs to be avoided first. Ship S2, with the second-highest risk, is close to S1 in its current position, but its risk is slightly lower than S3 due to its higher speed and smaller relative velocity. S4 is a head-on ship. Although there is a distance from S1, the relative velocity is fairly high, and the collision risk reaches Class III. Finally, S5, the farthest ship, has the lowest risk. In general, the ship does not need to avoid a collision when the risk level is Class IV. However, the ship in Class IV has a potential hazard. Therefore, as discussed in Section 3.2, the potential hazard still needs to be considered when making collision-avoidance decisions.

## 3. Improvement of Velocity Obstacle Algorithm

### 3.1. Avoiding Single Obstacles

Due to the different shapes and sizes of static obstacles in inland rivers, this paper uses the first classification method and then a grid map to model. Specific rules are as follows: Static obstacles, such as central bars, islands, and shoals, are outlined as polygons. The reefs or piers with uncertain shapes are outlined as circles. As mentioned in Section 2.1, the ship domain is a circle when facing static obstacles, so the decision algorithm uses the traditional VO algorithm, as shown in Figure 9.

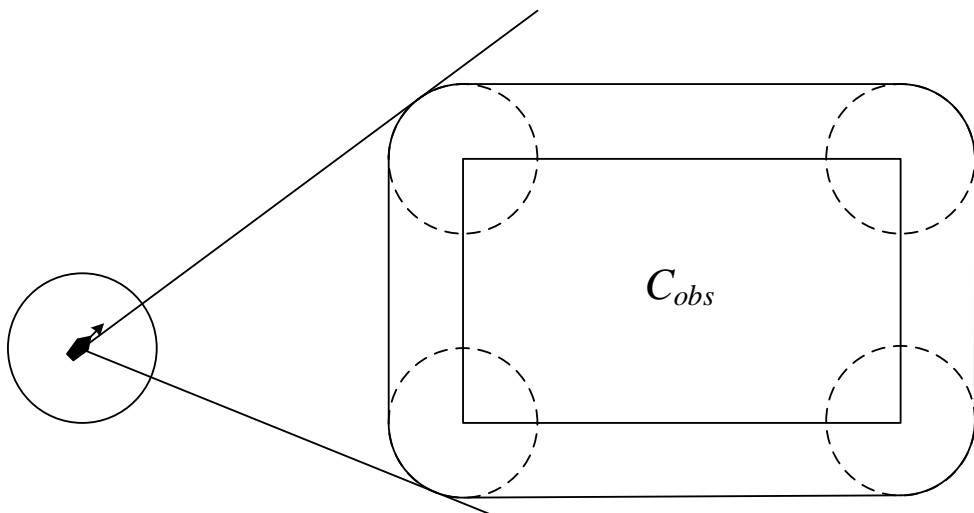

**Figure 9.** Avoidance of polygonal obstacles.

In Figure 9, $C_{obs}$ represents the obstacle domain.

Facing the dynamic obstacles, the left eccentric elliptical ship domain is adopted, and $C_{obs}$ cannot be established. For the obstacle-avoidance problem of irregular graphics, some scholars have transformed the irregular regions into regular regions, such as the quaternion domain, which transforms the collision part into a circle. Some scholars have also used

the grid method to solve the regional quantization, but there is a problem with too large data particles. The deviation of the avoidance angle calculated by the above two methods may cause the deviation of the avoidance point, which will lead to a failure of collision avoidance. In order to ensure accuracy, analytical geometry is used in this paper, as shown in Figure 10.

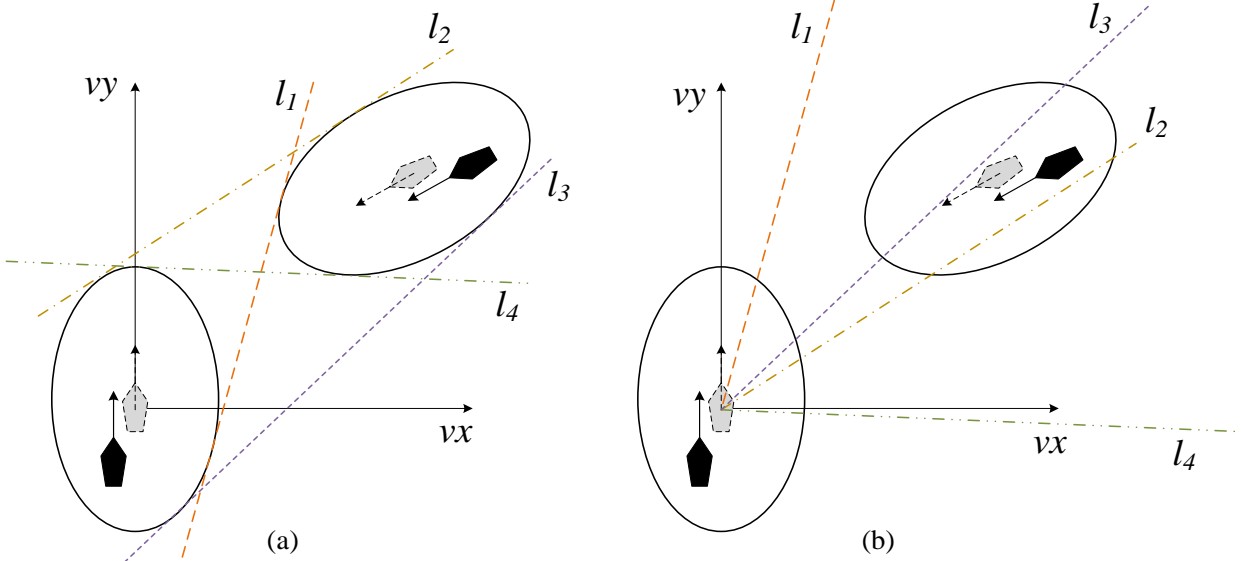

**Figure 10.** Avoidance in the elliptical ship domain: (**a**) determination of common tangents, (**b**) construction of VO cones.

In Figure 10a, there are four common tangents between two disjoint ellipses. Translate these tangents to the position where the virtual ship is located and select the tangents that form the maximum angle, as shown in Figure 10b. The area formed by $l_1$ and $l_4$ is the *RCC*.

*3.2. Multi-Ship Collision Avoidance*

In a multi-ship situation, the VO cone of each target ship should be merged and rezoned, and then the risk level should be added, as shown in Figure 11. Since the ships with Class IV are considered potential hazards in Section 2.3, their risk cannot be ignored, although collision avoidance is not considered. Therefore, add Class IV to calculate the collision risk of the new VO cone.

In Figure 11, the three VO cones are represented by A, B, and C, in which VO cone B overlaps with VO cone C to form the new VO cone D. Case 1: If the collision risks of S2 to S5 are 6.0, 5.5, 3.0, and 5.0, respectively, and the corresponding risk levels are Classes III, III, IV, and III, the collision risk of VO cone A is 9.0, and that of VO cone D is 10.5. Therefore, S1 should give priority to S3 and S5. Case 2: If the collision risks of S2 to S5 are 6.0, 5.0, 4.0, and 4.5, respectively, and the corresponding risk levels are Classes III, III, IV, and III, the collision risk of VO cone A is 10.0, and that of VO cone D is 9.5. Therefore, S1 should give priority to S2.

It should be noted that the turning time of inland ships is usually 1 to 3 min. After the collision risk is detected by the system very quickly, the ship has sufficient time to observe the behavior of the target ship. Therefore, it is not necessary to avoid obstacles and change course immediately. Therefore, this paper considers the safety distance, which is set as 6 L considering the width of inland channels and the size of ship domain. In general, it can be set smaller according to actual data. The crew can consider avoiding obstacles when the distance between the target ship and the own ship is close to a safe distance. Suppose this distance is greater than a safe distance. In that case, there is no need to make avoidance instructions immediately, and the own ship can maintain the current state to the next decision cycle.

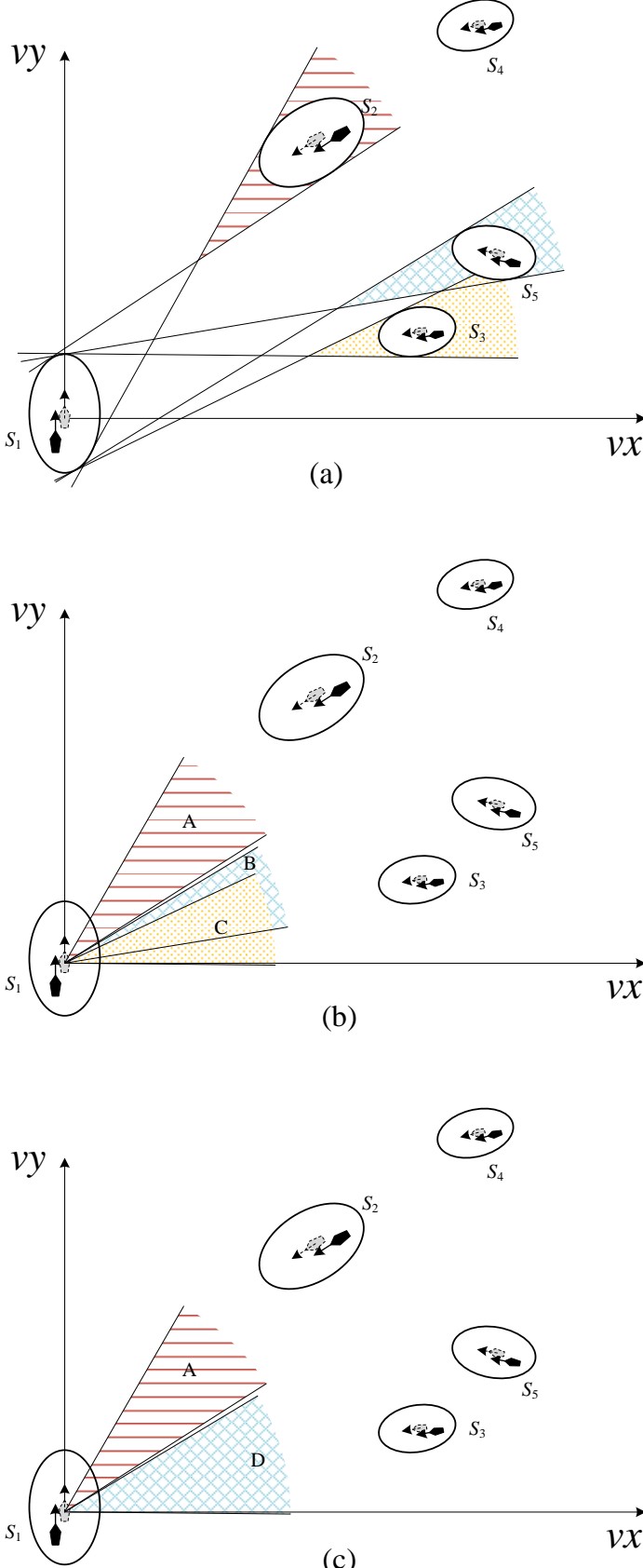

**Figure 11.** Combination of velocity obstacle cones: (**a**) determination of common tangents, (**b**) construction of VO cones, and (**c**) merging VO cones.

### 3.3. Collision Avoidance in Close-Quarters Situations

The method to avoid collision in close-quarters situations is establishing a speed buffer zone. When an obstacle ship suddenly changes its course and rushes into a speed buffer zone, the own ship will immediately enter the close quarters. In this state, the ship will avoid collision with a full rudder. To ensure safety and reduce loss as much as possible, it is even possible to abandon the inland-river collision-avoidance regulations. Collision avoidance in close-quarters situations is shown in Figure 12.

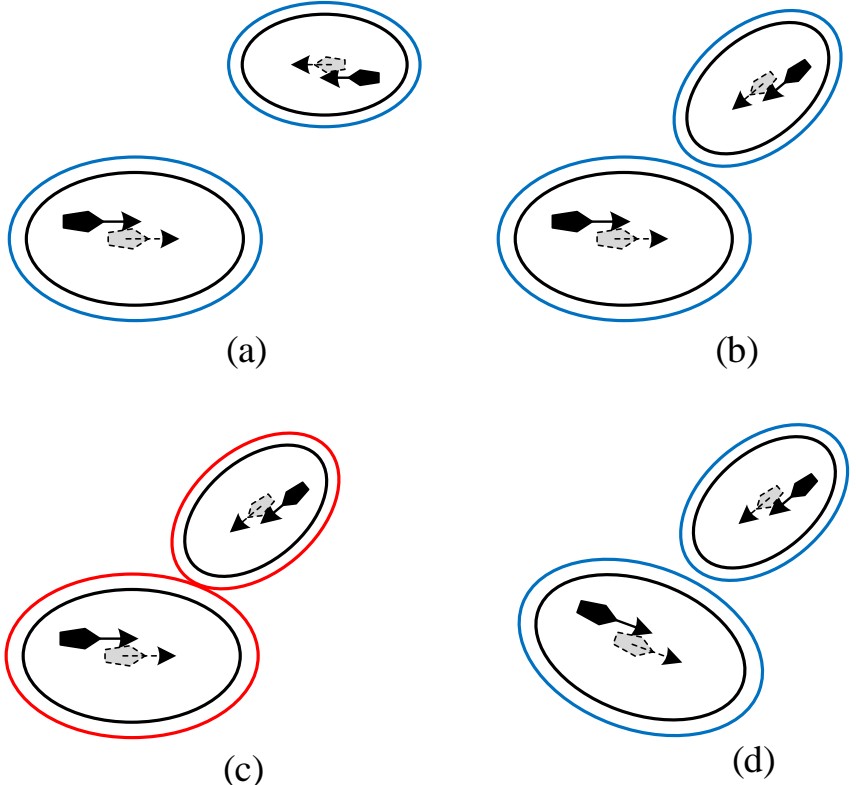

**Figure 12.** Collision avoidance in close-quarters situations: (**a**) scenario 1, (**b**) scenario 2, (**c**) scenario 3, and (**d**) scenario 4.

In Figure 12a,b, the countercurrent ship in another channel suddenly changes its course and sails toward the own ship, forming a crossing situation with the own ship. According to the avoidance regulations, the target ship should give way to the own ship at this time. In Figure 12c, the target ship is tangent to the speed buffer zone of the own ship, forcing the own ship into a close-quarters situation. In Figure 12d, the own ship abandoned the avoidance regulations and turned away from the close-quarters situation.

### 3.4. Collision-Avoidance Decision System Algorithm Flow

The flow chart of the inland-ship collision-avoidance decision system is shown in Figure 13.

As shown in Figure 13, if the risk is less than Class IV, the ship does not need to consider collision avoidance and can maintain its current navigation status. When the risk level is greater than Class IV, it indicates a collision risk, and the ship enters the collision-avoidance state. If the risk is greater than Class III, the ship enters a close-quarters situation and turns at the full rudder. Since the design speed for inland ships is generally between 18 km/h and 30 km/h, the time step can be 0.1 s. Finally, the collision-avoidance decision-making system outputs a rudder angle command to the crew or the control system of the USV to complete the collision-avoidance decision task.

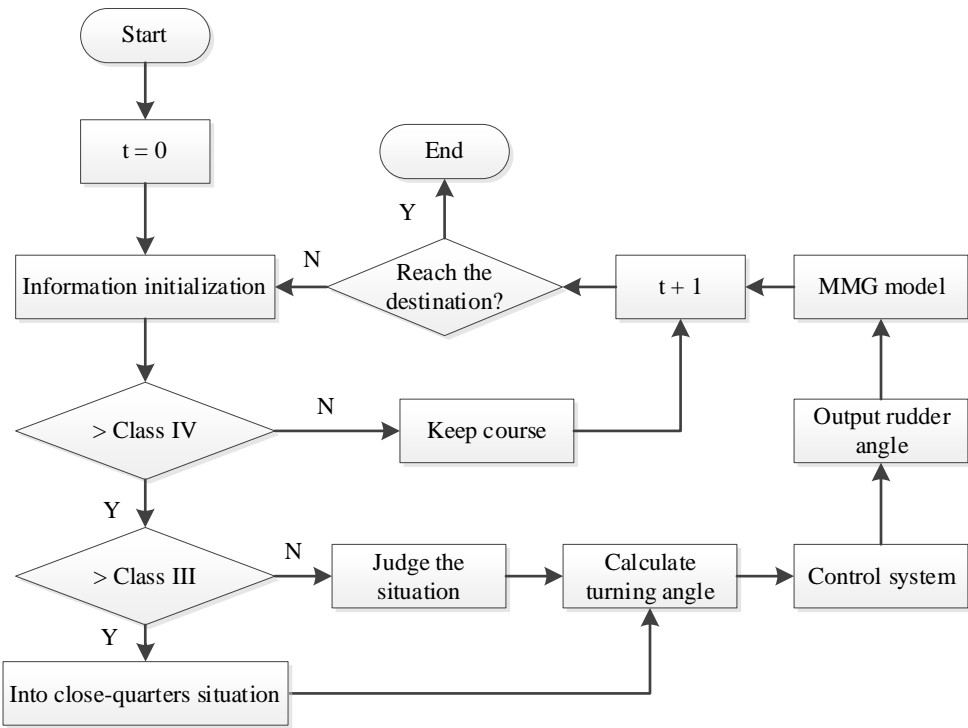

**Figure 13.** Flow chart the of collision-avoidance decision system.

## 4. Results

This paper uses MATLAB software to build a simulation platform. In the platform, the unit length of the map is 100 m. The abscissa points east and the ordinate points north. The simulation results and analysis are as follows.

### 4.1. Simulation Results of Static-Obstacle Avoidance

The simulation result of static-obstacle avoidance is shown in Figure 14.

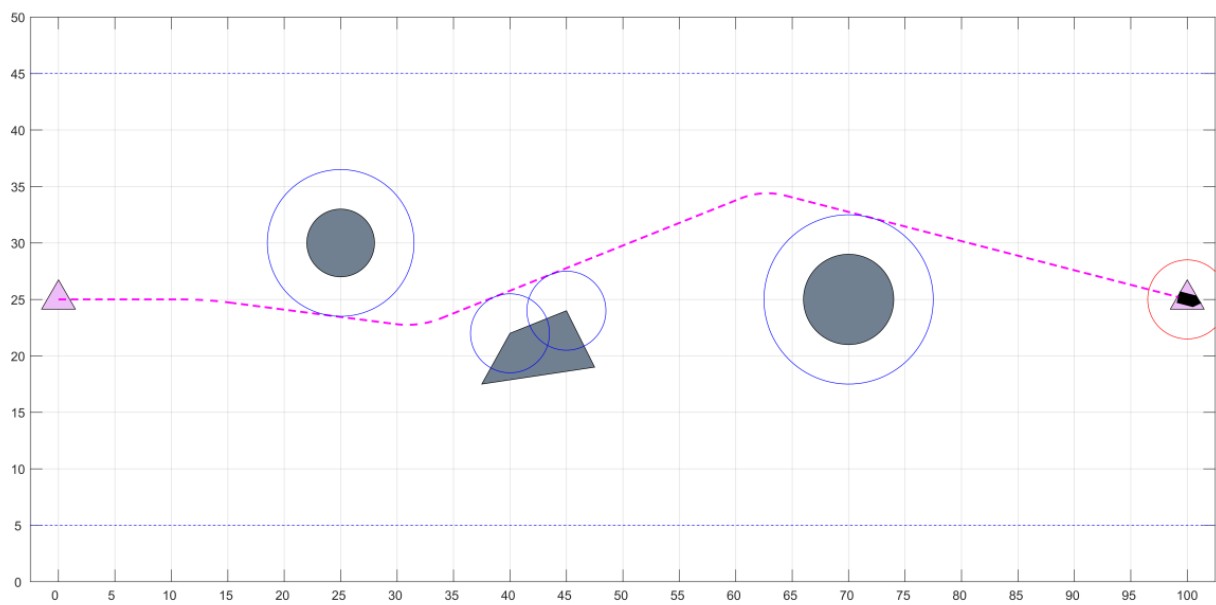

**Figure 14.** Simulation result of static-obstacle avoidance.

As shown in Figure 14, the ship sails from the starting point [0, 25] to the endpoint [100, 25]. The red circle is the ship domain, and the pink dotted line is the ship trajectory.

The ship needs to face two circular obstacles and one polygonal obstacle on its way. The blue circle on the obstacle is used to build $C_{obs}$. It can be seen that the ship's trajectory is tangent to $C_{obs}$, and the obstacles do not invade the ship's domain, so obstacle avoidance is effective.

### 4.2. Simulation Results of Multi-Ship Encounter Situation

This paper selects four ships to verify collision avoidance in a multi-ship situation, as shown in Table 6 and Figure 15.

**Table 6.** Ship initial information.

| No. | Starting Point (km) | Endpoint (km) | Speed (km/h) | Heading (°) |
|---|---|---|---|---|
| S1 | [0, 5.3] | [20, 5.3] | 32 | 90 |
| S2 | [2.5, 5.3] | [1.8, 5.3] | 24 | 90 |
| S3 | [9.6, 1.0] | [4.0, 8.0] | 20 | 45 |
| S4 | [16.5, 8.0] | [6.0, 0.8] | 18 | 270 |

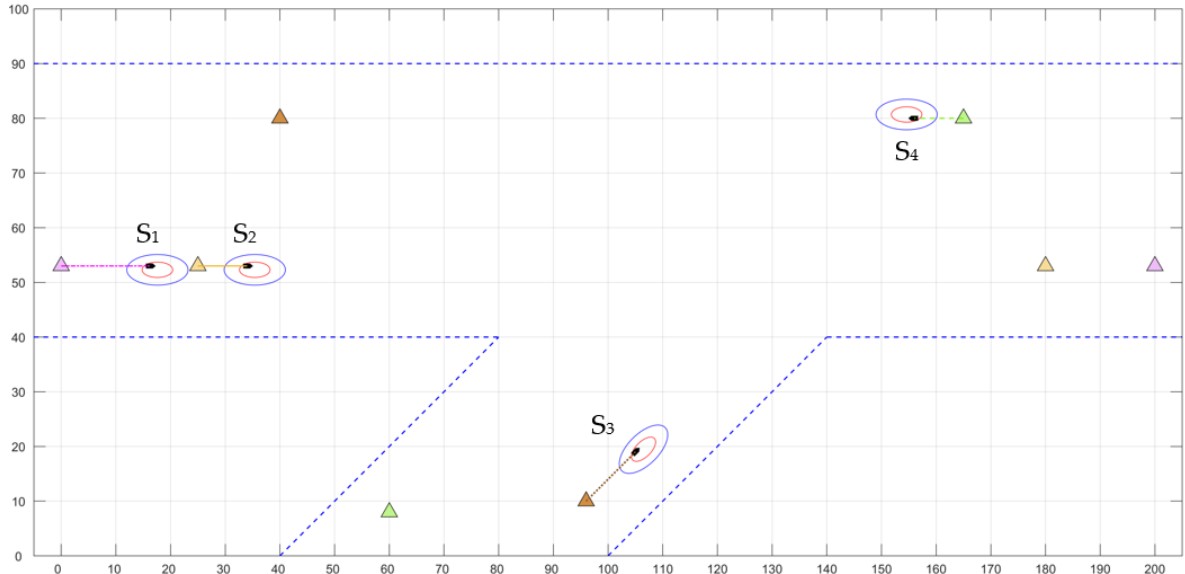

**Figure 15.** Verification of collision avoidance in a multi-ship situation.

As shown in Figure 15, there are four inland ships, S1, S2, S3, and S4, sailing to their respective target points. S1 and S2 form an overtaking situation in the down channel of the main river. S4 in the up channel wants to sail to the branch river, and S3 in the branch river intends to sail to the main river.

As shown in Figure 16, S1 is overtaking S2 on the starboard side. The avoidance point is [18.77, 53.10], and the avoidance angle is 10.23°. At this time, S3 is about to enter the main river.

Figure 17 shows that S1 has overtaken S2 and returned to the scheduled route. The turning point is [54.80, 59.46], and the turning angle is 14.94°. S3 and S4 form a crossing situation. According to the avoidance regulations, S3, as a give-way ship is passing the stern of S4. The avoidance point is [118.41, 43.44], and the avoidance angle is 53.29°. At this time, S4 and S2 form a crossing situation. S4 is a give-way ship and needs to give way from the stern of S2. The avoidance point is [126.65, 74.31], and the avoidance angle is 16.91°.

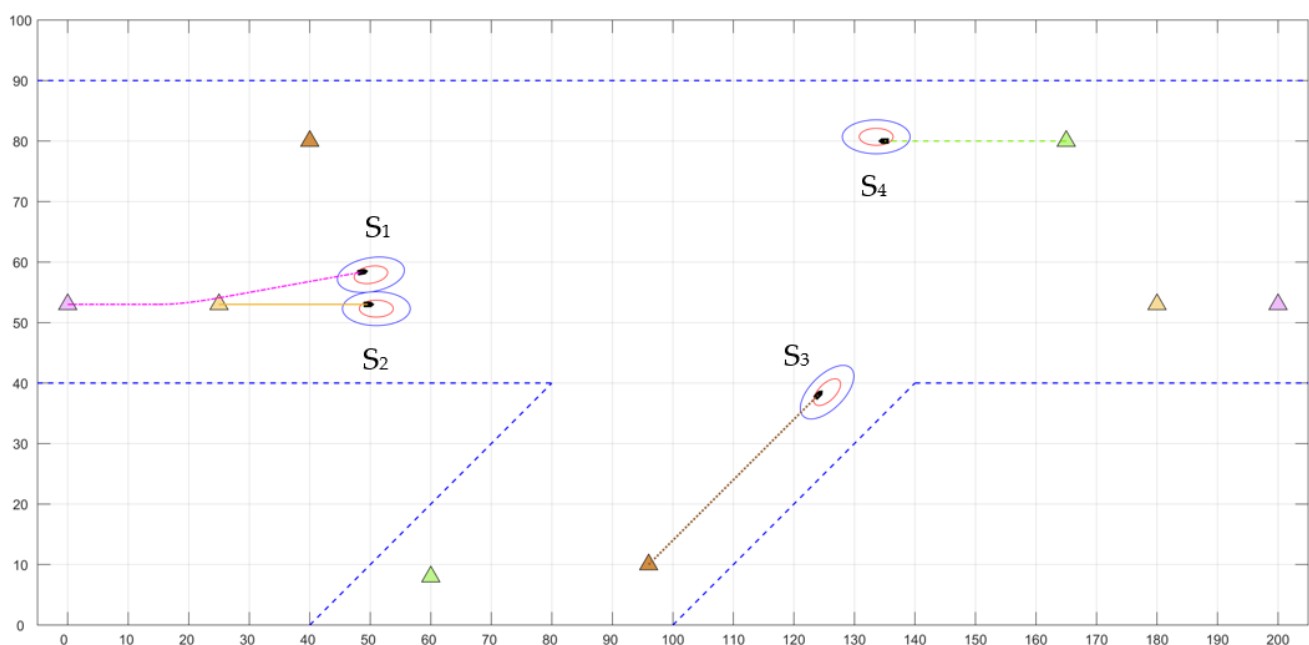

**Figure 16.** Scenario I of collision avoidance in a multi-ship situation.

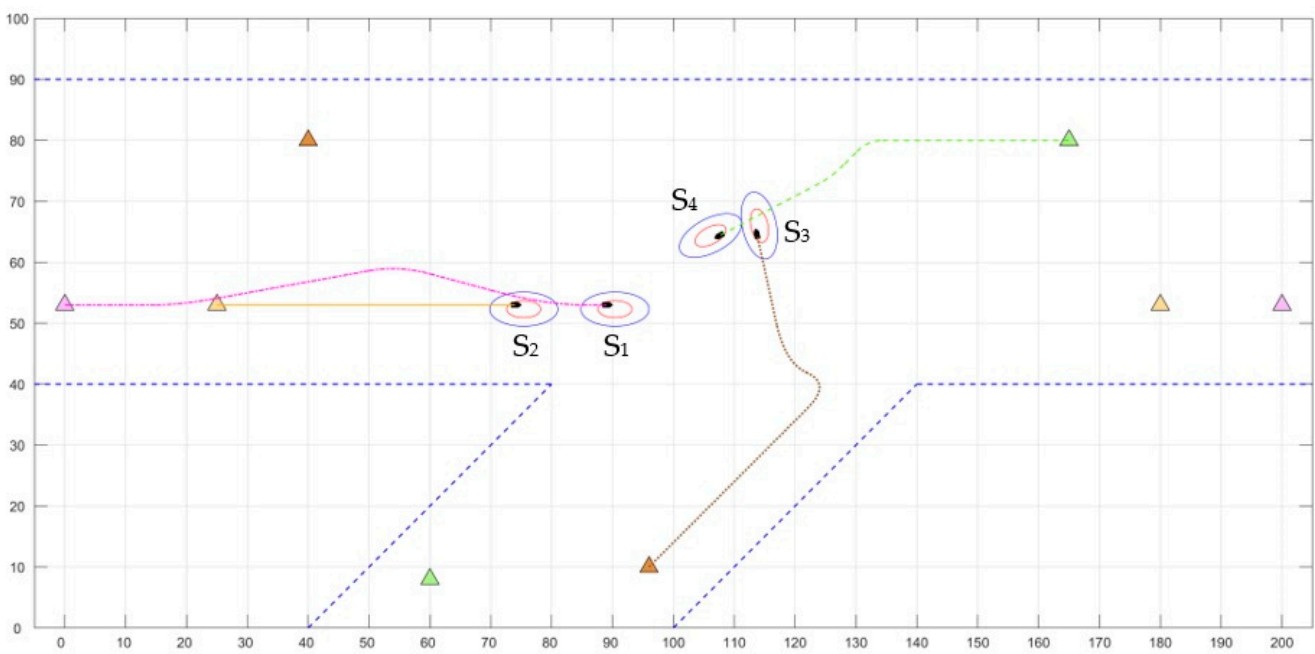

**Figure 17.** Scenario II of collision avoidance in a multi-ship situation.

Figure 18 shows that S3 has passed S4 and continues to sail to the target point. The turning point is [112.81, 68.44], and the turning angle is 67.95°. The give-way ship S4 is passing the stern of S2. The final collision-avoidance trajectory is shown in Figure 19.

From the above simulation, it can be seen that the four inland-river ships have successfully avoided collision, and have obtained a reasonable avoidance trajectory. The simulation results show that the inland-river collision-avoidance decision-making system designed in this paper is reasonable and effective. This system enables inland ships to deal with various encounter situations and make safe avoidance under the conditions of avoidance regulations and maneuverability restrictions.

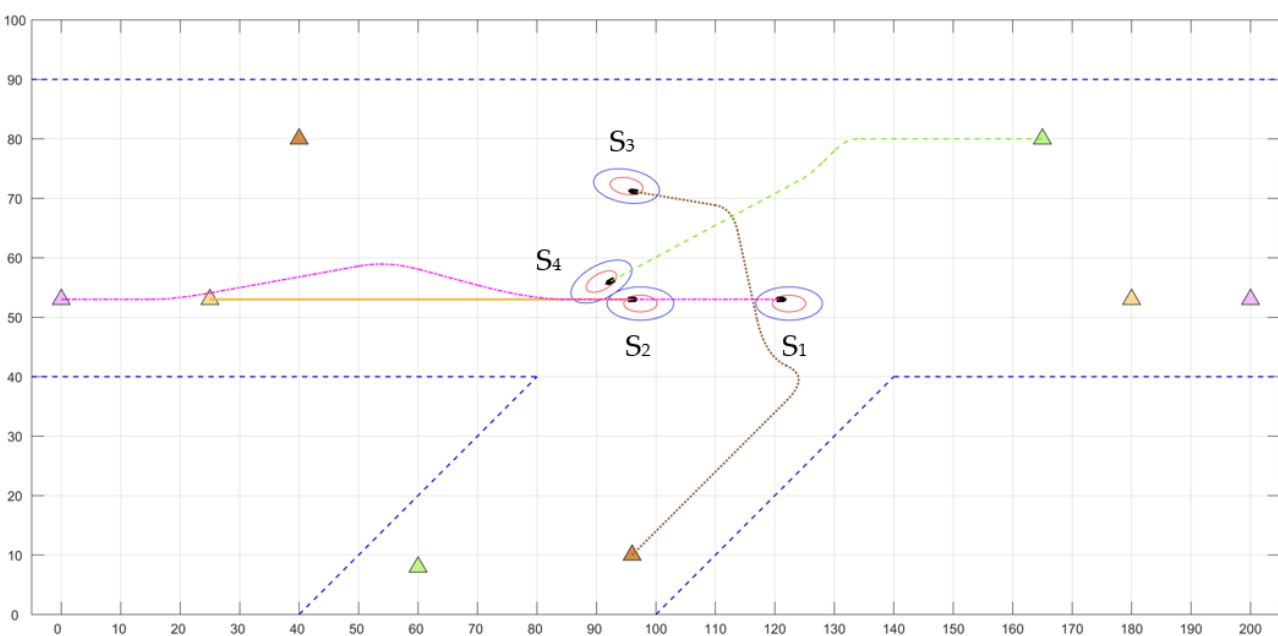

**Figure 18.** Scenario III of collision avoidance in a multi-ship situation.

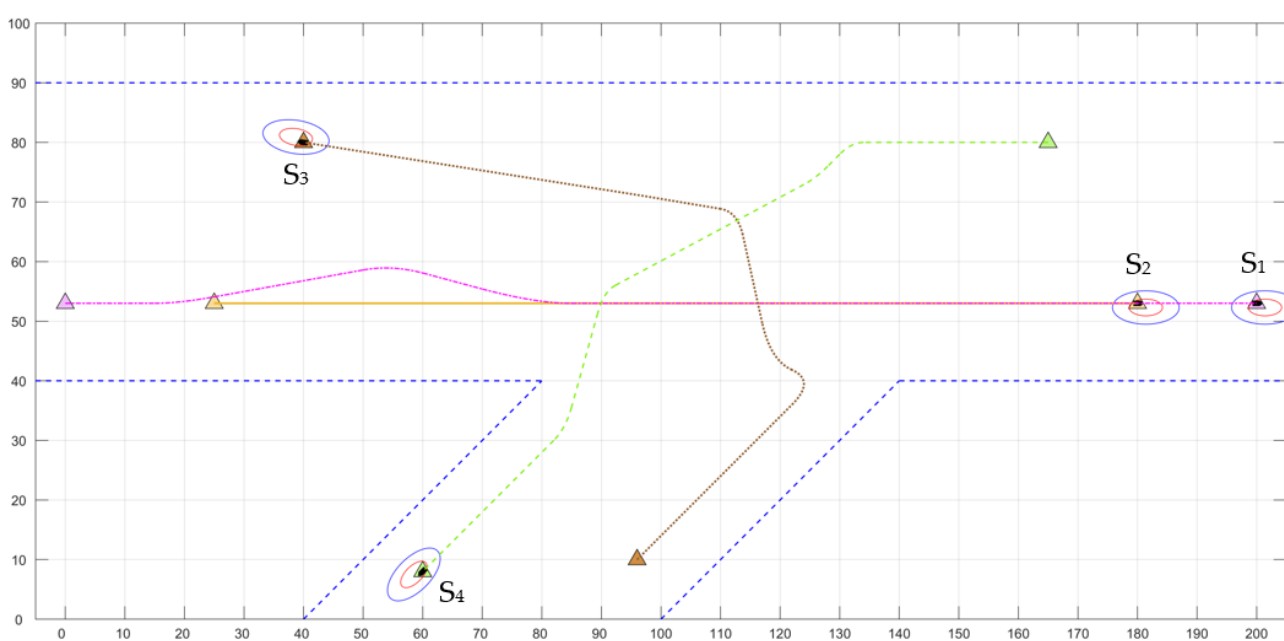

**Figure 19.** Multi-ship collision-avoidance trajectory.

*4.3. Simulation Results of a Close-Quarters Situation*

The simulation of the close-quarters situation is shown in Figure 20.

As shown in Figure 20a, the two ships are sailing safely in their respective channels, with S1 speed of 27 km/h and S2 speed of 24 km/h. Figure 20b shows that the ship below suddenly turns and invades the speed buffer zone. The turning angle is 42.30°. In Figure 20c, the ship above abandoned the inland-river avoidance regulations and turned left to avoid a collision. The avoidance point is [47.13, 27.51], and the avoidance angle is 60.40°. Figure 20d shows that S1 has completed avoidance, and the ship domains are not invaded. Figure 20e shows that the two ships have recovered the speed buffer zone after avoidance. Figure 20f shows that the two ships have broken away from the close-quarters situation, and returned to the scheduled route. S1 returned at the point [57.26, 33.18]. From the simulation results, it can be seen that the collision-avoidance system designed

in this paper can deal with the close-quarters situation and provide inland ships with an avoidance decision under the close-quarters situation.

**Figure 20.** Collision avoidance in close-quarters situation: (**a**) scenario 1, (**b**) scenario 2, (**c**) scenario 3, (**d**) scenario 4, (**e**) scenario 5 and (**f**) scenario 6.

## 5. Conclusions

This paper designs a decision-making system for inland-ship collision avoidance based on the velocity obstacle algorithms. The system uses the shallow-water-ship-maneuverability motion model to simulate the movement of inland ships, and then gives the collision-avoidance decision through the improved VO algorithms. The advantage is that, in this system, each inland ship can respond to static obstacles, dynamic obstacles, and urgent situations in real time according to the inland-river environment without coordinating with other ships. Moreover, the decision-making time is less than 0.1 s.

The decision-making system was simulated and tested by MATLAB software. The simulation platform worked perfectly as the trajectory was planned as desired. Each ship can complete collision avoidance independently following the inland-river collision-avoidance regulations. The simulation results demonstrate the effectiveness and feasibility of using the VO algorithm as a collision-avoidance decision-making algorithm for inland-ship collision problems, especially in dealing with close-quarters situations.

However, there are still many deficiencies in the research. Areas to be improved include the combination of collision avoidance at bridge areas and river bends, consideration

of ship-to-ship interaction, especially the effect of large ships on small ships, consideration of the angle and time of navigation restoration, and the introduction of control systems and real map data to create a perfect inland-ship collision-avoidance decision system.

**Author Contributions:** Conceptualization, G.Z. and H.W.; methodology, G.Z.; software, G.Z.; validation, G.Z., H.W. and Y.W.; formal analysis, G.Z.; investigation, G.Z.; resources, G.Z.; data curation, J.L. and W.C.; writing—original draft preparation, G.Z.; writing—review and editing, G.Z. and H.W.; visualization, G.Z.; supervision, H.W.; project administration, H.W.; funding acquisition, H.W. All authors have read and agreed to the published version of the manuscript.

**Funding:** This research was funded by National Natural Science Foundation of China, grant number U1964202.

**Institutional Review Board Statement:** Not applicable.

**Informed Consent Statement:** Not applicable.

**Data Availability Statement:** Not applicable.

**Conflicts of Interest:** The authors declare no conflict of interest.

## Nomenclature

| | | | |
|---|---|---|---|
| $ACC$ | absolute collision cone | $r$ | yawing angular velocity |
| $A_f$ | positive projection area above the waterline | $TCPA$ | shortest encounter time |
| $A_s$ | side projection area | $t_P$ | thrust deduction factor |
| $a_1, a_2, a_3, n$ | constant coefficient | $U$ | membership function |
| $a_H$ | flow-increasing factor | $U_{CRI}$ | collision risk |
| $B$ | ship width | $U_r$ | relative wind speed |
| $C_b$ | block coefficient | $u$ | longitudinal velocity |
| $C_{obs}$ | obstacle domain | $V$ | ship speed |
| $C_{wx}(a_R),$ $C_{wy}(a_R), C_{wn}(a_R)$ | wind pressure coefficient | $V_r$ | relative speed of the ship to water |
| $D$ | ship distance | $v$ | lateral velocity |
| $DCPA$ | shortest encounter distance | $v_r$ | lateral velocity of $V_r$ |
| $D_{deep}$ | estimation formula of deep water | $\vec{v}_o$ | own ship's velocity vector |
| $D_{shallow}$ | estimation formula of shallow water | $\vec{v}_t$ | target ship's velocity vector |
| $d$ | average ship draft | $w_P$ | wake factor |
| $E$ | external force | $X$ | hydrodynamic force of surging |
| $f(\lambda)$ | water depth correction function | $x_H$ | the distance from the fluid force point to the ship's center of gravity |
| $H$ | bare hull force | $Y$ | hydrodynamic force of swaying |
| $h$ | water depth | $Y'_v, Y'_{vv}, Y'_{vr},$ $N'_v, N'_r$ | hydrodynamic coefficient |
| $K$ | ship speed ratio | $\beta$ | drift angle |
| $L$ | ship length | $\delta$ | rudder angle |
| $N$ | hydrodynamic moment of yawing | $\varphi$ | heading angle |
| $o - xyz$ | motion coordinate system | $\varphi_o$ | own ship's heading angle |
| $o_o - x_o y_o z_o$ | fix coordinate system | $\varphi_t$ | target ship's heading angle |
| $P$ | propeller force | $\gamma$ | flow-straightening factor |

| $Q$ | the distance from ship's center to the bank | $\lambda$ | water depth draft ratio |
|---|---|---|---|
| $Q_1$ | the nearest avoidance distance from the bank | $\theta_t$ | relative bearing angle |
| $Q_2$ | the safe avoidance distance from the bank | $\rho$ | water density |
| $R$ | rudder force | $\rho_a$ | air density |
| $RCC$ | relative collision cone | | |

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
