# Peer review of "Collision-Avoidance Decision System for Inland Ships Based on Velocity Obstacle Algorithms"

_jmse, doi:10.3390/jmse10060814_

Round 1
Reviewer 1 Report
I send the review in the attachment.

Author Response
Dear Reviewer,
Thank you for giving us important comments as shown below. These comments are all valuable and very helpful for revising and improving our paper, as well as the important guiding significance to our researches. Authors confirmed following comments and made our responses as shown below.
Point 1: I propose to extend the literature in the first paragraph of the introduction, related to the importance of a collision avoidance decision system for inland manned and unmanned ships that are on the Journal Citations Reports (JCR) list such as for example: …
Response 1: The author has added the description about the articles you provided to the first paragraph of the Introduction and References, as follows.
'In addition, in recent years, more and more inland unmanned surface vehicles (USV) have been applied. Giordano et al. have described a prototype of a marine drone optimized for very shallow water, which enables bathymetric surveys to be performed in areas that are not feasible for traditional boats [1]. Stateczny et al. completed the study on the positioning accuracy of GNSS/INS systems supported by DGPS and RTK receivers for hydrographic surveys by using the USV [2]. Nikolakopoulos et al. analyzed USV data to investigate the beach rock formations [3]. Specht et al. proposed the concept of an innovative autonomous unmanned system for shallow water depth monitoring [4]. According to the problem in the USV's navigation mode without the satellite navigation system, Wang et al. proposed the multi-sensor integrated navigation method of INS/CNS/DVL using AISFF [5]. It can be seen that inland USV also has the problem of collision avoidance in restricted waters or shallow waters. To solve this problem, …'
Point 2: All horizontal and vertical axes of the figures should be described. Moreover, SI units have to be considered.
Response 2: Thank you for your important comment. For the figures that introduce VO algorithms, such as Figure 1, Figure 10 and Figure 11, the author has added the description, as follows. 'the vx-axis points east, the vy-axis points north, …'
The meanings and parameters of other coordinate axes are described in Paragraph 2 Section 2.2 and Table 1. On this basis, the author checked all coordinate systems and confirmed that they use the SI units.
Special thanks to you for your important comments and suggestions.

Reviewer 2 Report
1) The Abstract is mainly descriptive. In the Abstract the Authors should add some of the most important results obtained in this research (its exact values). Such addition will highlight the novelty of the presented paper already in the Abstract. Therefore, the Abstract requires re-arrangement and addition of the most important obtained results.
2) In the paper should be added a Nomenclature inside which will be listed and explained all abbreviations, symbols and markings used throughout the paper text. Sometimes, the reader is required to turn back through the paper to find the proper meaning of some abbreviation, symbol or marking – if all of them are placed in the Nomenclature (in one place), then will be no difficulties during paper reading. The Nomenclature will notably improve reading experience (at least in my opinion).
3) Line 88 – this sentence cannot start with “where”. It should be written, for example, “In Figure 1, v0 is…” or something similar. The same should be corrected in Line 162, Line 338 and anywhere else in the paper (if occur).
4) Figure 7 – the differences between real ship and simulation should be further explained and discussed. Are they acceptable? What is the acceptable accuracy and precision range, etc.? The Authors have stated that: “The ship model can roughly capture the maneuvering motions…” – the question which should be explained and discussed is how rough the discrepancy can be for considering it as still acceptable?
5) Line 377 - the safety distance is set to 6L. The Authors should explain and discuss this exact selection. Why safety distance is not larger (or shorter) and how any other selection will effect results obtained in this research?
6) Subsections 3.3 and 3.4 cannot have exactly the same title. The title of subsection 3.4 should be: Collision avoidance decision system algorithm (or similar, but a title correction is surely required).
7) In the Introduction the Authors have stated: “Compared with previous ship collision avoidance algorithms, the operation speed is greatly improved.”. In the paper is not provided any result and evidence which will confirm this statement. Therefore, the Authors are required to add evidences and discussion related to the mentioned fact, because this should be one of the most important advantages of developed algorithm (in comparison to other algorithms).
8) In the paper, the Authors should add direct comparison (or at least a discussion) of the benefits which provide the developed algorithm (system) and which are missing in other similar algorithms or systems.
9) As the Abstract, the Conclusions section should also be improved with the most important obtained results (its exact values). Also the Conclusions seem to be too descriptive and general, without any details obtained in the presented analysis.
Final remarks: This is very interesting and innovative article, but it should be carefully and properly improved (according to the comments above) before it can be considered for publication.
Author Response
Dear Reviewer,
Thank you for giving us important comments as shown below. These comments are all valuable and very helpful for revising and improving our paper, as well as the important guiding significance to our researches. Authors confirmed following comments and made our responses as shown below.
Point 1: The Abstract is mainly descriptive. In the Abstract the Authors should add some of the most important results obtained in this research (its exact values). Such addition will highlight the novelty of the presented paper already in the Abstract. Therefore, the Abstract requires re-arrangement and addition of the most important obtained results.
Response 1: Follow your advice, the author has updated the Abstract, as follows.
‘Due to the complex hydrology and narrow channel of inland rivers, ship collision accidents occur frequently. The traditional collision avoidance algorithms are often aimed at the sea areas, but little at the inland rivers. To solve the problem of inland ship collision avoidance, this paper proposes an inland ship collision avoidance decision system based on the velocity obstacle algorithm. The system is designed to assist ships in achieving independent collision avoidance operations under the limitation of maneuverability while meeting inland ship collision avoidance regulations. First, the paper improved Maneuvering Modeling Group (MMG) model suitable for inland rivers. Then, this paper improves velocity obstacle algorithms based on the dynamic ship domain, which can deal with different obstacles and three encounter situations (head-on, crossing, and overtaking situations). In addition, this paper proposes a method to deal with close-quarters situations. Finally, the simulation environment built by MATLAB software is used to simulate the collision avoidance of inland ships against different obstacles under the different situations with the decision-making time less than 0.1s. Through the analysis of the simulation results, the effectiveness and practicability of the system are verified, which can provide reasonable collision avoidance decisions for inland ships.’
Point 2: In the paper should be added a Nomenclature inside which will be listed and explained all abbreviations, symbols and markings used throughout the paper text. Sometimes, the reader is required to turn back through the paper to find the proper meaning of some abbreviation, symbol or marking – if all of them are placed in the Nomenclature (in one place), then will be no difficulties during paper reading. The Nomenclature will notably improve reading experience (at least in my opinion).
Response 2: Thank you for your important reminding. The author has added the Nomenclature after the References.
Point 3: Line 88 – this sentence cannot start with “where”. It should be written, for example, “In Figure 1, v0 is…” or something similar. The same should be corrected in Line 162, Line 338 and anywhere else in the paper (if occur).
Response 3: Thank you for your reminding. The author has modified them.
Point 4: Figure 7 – the differences between real ship and simulation should be further explained and discussed. Are they acceptable? What is the acceptable accuracy and precision range, etc.? The Authors have stated that: “The ship model can roughly capture the maneuvering motions…” – the question which should be explained and discussed is how rough the discrepancy can be for considering it as still acceptable?
Response 4: Generally, the differences between real ship and simulation have no exact accuracy standard. In Figure 7, it can be seen that within the first 100s, the heading angle difference between the simulation data and the real ship data is about 1°, and the time difference is within 2s. This proves that the simulation data can restore the motion state of the real ship in a reasonable time. Since the Z-type test will hardly occur in the real sailing, the change of heading angle is consistent within the time shown in the Figure 7, which can prove the effectiveness of the ship model.
The author has added explanation in the last paragraph of Section 2.2, as follows.
‘Through comparison, it can be seen that within the first 100s, the heading angle difference between the simulation data and the real ship data is about 1°, and the time difference is within 2s. It is evident that the heading angles of the two groups of data are almost consistent with the change of rudder angle. This proves that the ship model can roughly capture the maneuvering motions in a reasonable time and is useful for the maneuvering predictions of inland ships.’
Point 5: Line 377 - the safety distance is set to 6L. The Authors should explain and discuss this exact selection. Why safety distance is not larger (or shorter) and how any other selection will effect results obtained in this research?
Response 5: Considering the width of inland channels and the size of ship domain, the author sets the safety distance as 6L, which can make sure the two ships as safe as possible. In general, it can be set smaller.
In addition, the length of the safety distance does not affect the safety of the algorithm, because the system will give the avoidance angle outside the safe distance, and setting the safety distance can give the crew enough time to decide whether to avoid.
Follow your advice, the author has added the explanation about the safety distance, as follows.
‘..., which is set as 6L considering the width of inland channels and the size of ship domain. In general, it can be set smaller according to actual data.’
Point 6: Subsections 3.3 and 3.4 cannot have exactly the same title. The title of subsection 3.4 should be: Collision avoidance decision system algorithm (or similar, but a title correction is surely required).
Response 6: Sorry for the mistake. The author has modified it. The title of subsection 3.4 is ‘Collision avoidance decision system algorithm flow’.
Point 7: In the Introduction the Authors have stated: “Compared with previous ship collision avoidance algorithms, the operation speed is greatly improved.”. In the paper is not provided any result and evidence which will confirm this statement. Therefore, the Authors are required to add evidences and discussion related to the mentioned fact, because this should be one of the most important advantages of developed algorithm (in comparison to other algorithms).
Response 7: The advice involves the principle of the algorithm (Line 108-115). Previous collision avoidance algorithms, such as Ant Colony Algorithms, Genetic Algorithms, and Particle Swarm Optimization, often need to carry out a large number of iterations in advance to get the results. The running time also increases with the increase of the number of ships and the number of iterations, generally more than 5 s. Therefore, these methods are more suitable for global path planning, with the advantage that multiple obstacle avoidance paths can be obtained. While the VO algorithms used in this paper uses the geometric principle, with the advantage that it can deal with multi-objective obstacle avoidance problems in real time, but the solution is unique. In addition, this paper sets the time step to 0.1s (Line 435), which can be flexibly adjusted according to the actual situation.
The author has added the description about that, as follows.
‘Previous collision avoidance algorithms,..., often need to carry out a large number of iterations in advance to get the results. The running time also increases with the increase of the number of ships and the number of iterations, generally more than 5 s.’
‘The advantage is that, in this system, each inland ship can respond to static obstacles, dynamic obstacles and urgent situations in real time according to the inland river environment without coordinating with other ships. And the decision-making time is less than 0.1 s.’
Point 8: In the paper, the Authors should add direct comparison (or at least a discussion) of the benefits which provide the developed algorithm (system) and which are missing in other similar algorithms or systems.
Response 8: Thank you for your advice. The author has added the description, as follows.
‘Previous collision avoidance algorithms, such as Ant Colony Algorithms, Genetic Algorithms, and Particle Swarm Optimization, often need to carry out a large number of iterations in advance to get the results. The running time also increases with the increase of the number of ships and the number of iterations, generally more than 5 s. Therefore, these algorithms are more suitable for global path planning, with the advantage that multiple obstacle avoidance paths can be obtained. However, in case of a close-quarters situation, once own ship is unable to coordinate with the other ship immediately, the collision avoidance opportunity will be delayed, resulting in the accident. Based on geometric principles, the VO algorithm has a small amount of calculation and uniform results, which meets the basic requirements of the inland ship collision avoidance algorithm. Notably, the ship can make decisions independently. This allows the ship to receive immediate collision avoidance instructions in case of a close-quarters situation. In addition, in the inland rivers, multi-ship encounter situations often occur with the number of ships more than ten. For this problem, some scholars have used VO algorithm to complete the obstacle avoidance simulation test of 1000 agents [32], which is difficult for other algorithms.’
Point 9: As the Abstract, the Conclusions section should also be improved with the most important obtained results (its exact values). Also the Conclusions seem to be too descriptive and general, without any details obtained in the presented analysis.
Response 9: The author has updated the description of the Conclusions section. In the updated description, the author deleted redundant content and emphasized the algorithm advantages, as follows.
‘This paper designs a decision-making system for inland ship collision avoidance based on the velocity obstacle algorithms. The system uses the shallow water ship maneuverability motion model to simulate the movement of inland ships, and then gives the collision avoidance decision through the improved VO algorithms. The advantage is that, in this system, each inland ship can respond to static obstacles, dynamic obstacles and urgent situations in real time according to the inland river environment without coordinating with other ships. And the decision-making time is less than 0.1s.
The decision-making system is simulated and tested by MATLAB software. The simulation platform worked perfectly as the trajectory was planned as desired. Each ship can complete collision avoidance independently following the inland river collision avoidance regulations. The simulation results demonstrate the effectiveness and feasibility of using the VO algorithm as a collision avoidance decision-making algorithm for inland ship collision problems, especially in dealing with close-quarters situations.
However, there are still many deficiencies in the research. Areas to be improved include the combination of collision avoidance at bridge areas and river bends, consideration of the ship to ship interaction, especially the effect of large ships on small ships, consideration of the angle and time of navigation restoration, and the introduction of control systems and real map data to create a perfect inland ship collision avoidance decision system.’
Special thanks to you for your important comments and suggestions.

Round 2
Reviewer 2 Report
The Authors have performed all proposed corrections/additions. The additional explanations provided in the answers to my comments were very helpful for a full and complete understanding of all the details.
Now, after revision, I have no more concerns related to this paper. The paper should be published in a presented (revised) form.
My congratulations to the Authors.